# Carbohydrate Metabolism Parameters of Adult Glial Neoplasms According to Immunohistochemical Profile

**DOI:** 10.3390/biomedicines10051007

**Published:** 2022-04-27

**Authors:** Larisa Obukhova, Olga Nikiforova, Claudia Kontorshchikova, Igor Medyanik, Natalya Orlinskaya, Artem Grishin, Michael Kontorshchikov, Natalya Shchelchkova

**Affiliations:** Federal State Budgetary Educational Institution of Higher Education, Privolzhsky Research Medical University of the Ministry of Health of the Russian Federation, 603005 Nizhny Novgorod, Russia; nikolaevna160283@mail.ru (O.N.); kontclin@mail.ru (C.K.); med_neuro@inbox.ru (I.M.); orlinskaya_n@pimunn.ru (N.O.); patolog@pimunn.ru (A.G.); palkin2608@gmail.com (M.K.); n.shchelchkova@mail.ru (N.S.)

**Keywords:** kinase of glycogen synthase 3β, glucose-6-phosphate dehydrogenase, hexokinase, transketolase, peritumoral zone, molecular genetic markers of gliomas, IDH1, MGMT, Ki 67, p53

## Abstract

This research aimed to investigate the interrelationship of carbohydrate metabolism parameters and immunohistochemical characteristics of glial tumors. Tumor tissue, peritumoral area, and adjacent noncancerous tissue fragments of 20 patients with gliomas of varying degrees of anaplasia were analyzed. The greatest differences in the carbohydrate metabolism compared to adjacent noncancerous tissues were identified in the tumor tissue: reduction in the levels of lactate and glycogen synthase kinase-3β. Significant differences with adjacent noncancerous tissues for the peritumoral zone were not found. The activity of the carbohydrate metabolism enzymes was different depending on the immunohistochemical glioma profile, especially from Ki 67 level. Bioinformatic analysis of the interactions of immunohistochemical markers of gliomas and carbohydrate metabolism enzymes using the databases of STRING, BioGrid, and Signor revealed the presence of biologically significant interactions with glycogen synthase kinase 3β, hexokinase, glucose-6-phosphate dehydrogenase, and transketolase. The established interconnection of glycolysis with methylation of the promoter of O-6-methylguanine-DNA-methyltransferase (MGMT) of gliomas can be used to increase chemotherapy efficiency.

## 1. Introduction

Hyperglycemia is a risk factor that reduces survival in patients with glioblastomas [1]. High levels of insulin and insulin-like growth factor (IGF)-1, expressed under the action of carbohydrate-rich food, can contribute to the proliferation of tumor cells through the insulin signaling pathway/IGF1 [2,3,4]. K. Masur et al. [5] showed that a high concentration of glucose causes changes in the expression of genes activating tumor cell proliferation, migration, and adhesion. Adding insulin to the medium with a high content of glucose increased the proliferation rate by 20–40% and contributed to the activation of the PI3K pathway.

An important point is the study of carbohydrate metabolism, including in the peritumoral zone of the tumor. It is known that the peritumoral region of gliomas has a specific cellular composition and molecular features. It can perform both a limiting function and be a substrate for further tumor spread, depending on the stage of gliomagenesis and the predominant microglial phenotype. The infiltrative nature of the spread of gliomas into adjacent tissues is associated with the cross-exchange of cells in these areas, where extracellular microRNA vesicles are of the greatest importance [6]. Because of this, the study of carbohydrate metabolism in conjunction with other factors in this area may be useful to improve understanding of the role of the peritumoral zone in gliomas and predict their future behavior.

IDH mutations, methylation of MGMT promoter, p53 protein, and Ki-67 proliferative level [7] refer to the markers most frequently used in clinical practice determined by immunohistochemistry.

The isocitrate dehydrogenase gene (IDH) encodes the enzyme of the tricarboxylic acids cycle [8]. Isocitrate dehydrogenase 1, which is mainly located in the cytoplasm, catalyzes the oxidative decarboxylation of isocitrate to form alpha-ketoglutarate and the reduced form of NADPH_2_. IDH mutations appear in the earliest stages of tumor formation, participating in gliomagenesis [9].

p53 protein, a tumor suppressor, induces apoptosis during the damage to the genome and blocks the cell cycle preventing the accumulation of genetically defective cells [10]. TP53 mutations lead to the inactivation of the relevant protein and further development of oncogenesis [11].

Ki-67 is the nuclear protein associated with cell proliferation and it takes part in ribosomal RNA synthesis [12]. It is found in all active phases of the cell cycle and does not manifest itself in the resting phase. Its expression reflects the severity of the proliferative activity of the tissue. In glial neoplasms, Ki-67 is used as an auxiliary marker for differential diagnostics of Low Grade (I, II) and High Grade (III, IV) gliomas [13,14].

O-6-methylguanine-DNA-methyltransferase (MGMT) carries out DNA repair [15]. The binding of the methyl group to the promoter limits the ability of DNA-polymerase to connect with it, making further gene expression impossible [16]. Methylation of the MGMT promoter is the most significant predictive marker of the severity of an alkylating agent’s response to chemotherapy [17].

The purpose of this study was to evaluate the relationship of the parameters of carbohydrate metabolism with immunohistochemical characteristics of glial tumors.

## 2. Materials and Methods

### 2.1. Materials

Tumor tissue, peritumoral area, and their adjacent noncancerous tissues were collected as postoperative fragments at the Federal State Budgetary Educational Institution of Higher Education «Privolzhsky Research Medical University» of the Ministry of Health of the Russian Federation with informed consent prior to anticancer therapy. Histological diagnosis was made using the WHO classification of CNS tumors [7]. The detailed information of patients was listed in Appendix A.

### 2.2. Measurement of Tumor Markers

Before staining, the standard procedure for dewaxing and retrieval was carried out. Such antibody clones as Anti-IDH1 R132H (Dianova International, Sevilla, Spain), Anti-MGMT (EP337 clone), AC-0307RUO number (Epitomics, Burlingame, CA, USA), Anti-p53 (DO-7 clone) (Leica Biosystems, Heidelberg, Germany), and Ki-67 antibodies (SP6 clone) (Thermo Scientific, Waltham, MA, USA) were used. The presence of an IDH1 mutation was estimated by the presence of cytoplasmic expression. The level of markers MGMT, Ki-67, p-53 was calculated by the ratio of positive nuclear staining. In the study, the staining with only moderate or high severity was taken into account (Figure 1). The values of the extent of IDH1 and p53 were divided into groups: less than 1/10, from 1/10 to 1/3, from 1/3 to 1/2, and more than 1/2 of painted cells [18,19]. The calculation of Ki-67 and MGMT ratio was carried out in 10 fields of view with a magnification of ×400 [20]. The values of the results were expressed as the ratio of painted cells in 10 fields of view [21]. While studying the presence of methylation of the MGMT promoter, the nuclear staining with less than 15% of cells was considered to be a positive result [22].

Since IDH, MGMT, and p53 are estimated in the presence/absence format without measurement units, the results are presented as % of cases of marker presence. While evaluating Ki-67, the % staining in the preparation was determined, and, thus, for this marker, the results are indicated as Medians and Interquartile ranges.

### 2.3. Preparation of Tissue Homogenate for Biochemical Research

Biochemical studies were carried out in tissue homogenate. Preparation of tissue homogenate was carried out in the refrigeration room at 0 °C. The postoperative material was washed with a 0.32 M sucrose solution, pH = 7.4, and was separated from the shells. Then, the tissue was homogenized at the rate of 200 rpm in a homogenizer (glass-Teflon) in a 10-fold volume of the isolation medium containing 0.32 mol of sucrose, 10 mmol of Tris-HCl, and 1 mmol of EDTA, pH = 7.4.

### 2.4. Analysis of the Biochemical Parameters of Carbohydrate Metabolism

Determination of the concentration of carbohydrate metabolism metabolites was carried out calorimetrically using the UV-mini 1240 spectrophotometer, Shimadzu. For analysis of glucose concentration, a set of reagents Climates-GOT from SPC “Eco-Service”, Moscow, Russia (Cat No. B-11061) was used. The concentration of lactate was determined using the set from OOO «Olvex Diagnosticum», Russia (Cat. No. 019.002). The activity of glucose-6-phosphate dehydrogenase was determined using the set of reagents «Sentinel», Milano, Italy (Cat. No. 17005) on the biochemical analyzer StatFaks 04, the USA. The activity of the regulatory enzyme of glycolysis, hexokinase, was determined by staining the probe using the set of reagents from «Sigma-Aldrich» (St. Louis, MO, USA) (Cat. No. MAK091) on the MULTISCAN spectrophotometer (Thermo Scientific, Vantaa, Finland). The immunoenzyme method of «sandwich» type was used to determine the amount of transketolase (Human SimpleSep ELISA Kit “Abcam”, Waltham, MA, USA; Cat. No. ab187398) and glycogen synthase kinase 3β, (ELISA kit “Puda Scientific”, WUHAN, China; cat. No. PD-H7862E); optical density was registered at 450 nm on the MULTISCAN spectrophotometer (Thermo Scientific, Vantaa, Finland).

The results were corrected to 1 g of protein which was determined by the Lowry method with a set of reagents from OOO «Firma Syntacon»,Saint Petersburg, Russia.

### 2.5. Analysis of Protein–Protein Interactions

To identify the relationship between the above-mentioned immunohistochemical markers and the enzymes of carbohydrate metabolism, the protein–protein interactions were analyzed using the databases of STRING (STRING: functional protein association networks (string-db.org)), BioGrid (BioGRID | Database of Protein, Chemical, and Genetic Interactions (thebiogrid.org)), Signor (SIGNOR 2.0 (uniroma2.it)), and KEGG PATHWAY (https://www.kegg.jp accessed on 9 February 2022). The integrative scheme of the obtained protein–protein interactions was built using the “yEd Graph Editor” program.

### 2.6. Statistical Analysis

Statistical processing of the data was carried out using the AnalystSoft Inc. package, StatPlus, version 6, Alexandria, VA, USA (https://www.analystsoft.com accessed on 9 February 2022). The results were presented in the form of medians, percentiles, and quartiles (25%; 75%). The significance of the differences obtained was evaluated using non-parametric criteria (Mann–Whitney U-criterion, Kolmogorov–Smirnov criterion). Correlation analysis was carried out with the determination of Spearman’s coefficient.

## 3. Results

### 3.1. The Level of Molecular Genetic Markers of Gliomas with Varying Degrees of Anaplasia

The immunohistochemical profile was studied in the pilocytic astrocytoma grade I, in astrocytomas and oligodendrogliomas with the level of anaplasia II, III as well as in the primary glioblastomas, the midline glioma, and grade IV astrocytomas. The investigations were carried out on an IDH1 mutation as a prognostic marker of tumor behavior; on MGMT promoter methylation as a predictive response marker to chemotherapy; on Ki-67 level as a reflection of proliferative activity; on p53 protein as an indirect marker of astrocytic neoplasms included in the chain of oncogenesis. According to the results of immunohistochemical research, the IDH1 markers, methylation of the MGMT promoter, and p53 were found in all groups of gliomas (Table 1).

The value of the Ki-67 proliferative index increased with an increase in the degree of anaplasia and was the highest in the group of grade IV gliomas. The identification of the p53 protein was not explicitly dependent on the degree of malignancy (Table 1). However, the p53 tumor marker was absent in all four cases of oligodendrogliomas and was present in most astrocytic formations. The IDH1 mutation was traced in all cases of oligodendrogliomas and astrocytoma’s of grade II-III as well as in the secondary grade IV glioblastomas. Mutation of IDH1 was not detected in primary glioblastomas, midline glioma, and pilocytic astrocytoma. Methylation of MGMT promoter prevailed in II, III degree gliomas. The results obtained correspond to the classification of WHO for primary tumors of the CNS 2021.

### 3.2. Protein–Protein Interactions between Immunohistochemical Markers of Gliomas and the Main Enzymes of Carbohydrate Metabolism

To establish the existing interconnections between the immunohistochemical markers of gliomas and the carbohydrate metabolism enzymes, the bioinformatic analysis was carried out using the databases on the interaction of molecular biological objects. At the same time, the interactions are revealed between these markers of tumor growth of gliomas as IDH1 and TP53 and carbohydrate metabolism enzymes (kinase of glycogen synthase 3β, glucose-6-phosphate dehydrogenase, hexokinase 1/2, transketolase) (Figure 2). The data were obtained using STRING with an average confidence of 0.400. Protein–protein interactions were taken into account only if the combined score between the nodes was more than 0.6. Both indirect and direct effect of some proteins on the functionality of other proteins was revealed.

With the use of Signor, it was found that p53 protein reduces the activity of the G6PD enzyme. This data is confirmed by KEGG category Pathway, Human Diseases. In the metabolic map, containing the signaling pathway of carbohydrate metabolism in the case of carcinogenesis (Central Carbon Metabolism In Cancer), the inhibiting effects of p53 on G6PD are shown. It is also indicated in Signor that p53 reduces the activity of the regulatory glycolysis enzyme, HK2, and lowers the expression of MGMT.

Using the bioinformatics analysis, the effect of GSK3 β was revealed on more than 300 proteins due to their post-translation modification. The effect of immunohistochemical markers of gliomas on GSK3β activity was revealed. Thus, in Bio Grid the protein–protein interactions of TP53 and GSK3β with high-throughput interaction datasets were found. p53 protein, binding to GSK3β, activates the enzyme. Conversely, GSK3β phosphorylates tumor suppressor p53 and, thereby, activates it.

With the use of the databases on protein–protein interactions, the biologically significant interactions of IDH1 and p53 with glycogen synthase kinase 3β, hexokinase, glucose-6-phosphate dehydrogenase, transketolase are revealed.

### 3.3. Parameters of Carbohydrate Metabolism Depending on the Zone of Glial Tumor

To estimate the intensity of glycolysis processes, pentose phosphate pathway, and glycogen synthase kinase 3β in tumor tissues, the peritumoral zone, and their adjacent noncancerous tissues, the parameters of carbohydrate metabolism in the stated tissues were evaluated (Table 2).

Significant differences with adjacent noncancerous tissues in the studied carbohydrate metabolism parameters were detected for tumor tissue only (Table 2). The lactate content in the tumor was significantly lower (by 4.23 times) than in adjacent noncancerous tissues. The content of glycogen synthase kinase-3β was 2.87 times lower in the tumor zone than in adjacent noncancerous tissues.

### 3.4. Carbohydrate Metabolism Characteristics Depending on the Immunohistochemical Profile of Gliomas

The correlation dependence of carbohydrate metabolism enzymes with markers of tumor growth of gliomas was analyzed (Table 3). The greatest number of significant relationships was revealed for MGMT.

Further, the data on the parameters of the carbohydrate metabolism were divided into groups depending on the immunohistochemical profile accounting for the appropriate marker. The figures present only data with meaningful differences between groups.

In the group with mutations of IDH1 genes, the content of transketolase is higher in the tissue of the peritumoral zone and tumor (Figure 3). Therefore, it is possible to state that IDH1 mutations are accompanied by activation of the non-oxidative stage of the pentose phosphate pathway.

The level of transketolase in the peritumoral zone of the tumor was significantly higher for the IDH1 mutation group (U-criterion of Mann–Whitney *p* = 0.05; Kolmogorov–Smirnov criterion *p* = 0.1). This trend is observed both for adjacent noncancerous tissues and for tumor tissue, however, the difference between the Wild type IDH1 group is not significant.

In the group with p53 presence, the activity of the glucose-6-phosphate dehydrogenase in the tumor tissue was significantly lower (Figure 4) making it possible to prove the decrease in the rate of the oxidative phase of the pentose phosphate pathway.

Glucose-6-phosphate dehydrogenase activity in the tumor was significantly lower in the group with the expression of the tumor suppressor protein p53 (U-criterion of Mann–Whitney *p* = 0.04; Kolmogorov–Smirnov criterion *p* = 0.02) compared to the group without p53 expression.

With a high level of the mitotic index Ki-67, there is a lower activity of transketolase in the tissues of the tumor (significant) (Figure 5) which indicates the decrease in the activity of the oxidative stage of the pentose phosphate pathway.

The content of transketolase was significantly lower in the group with high mitotic index Ki-67 (U-criterion of Mann–Whitney *p* = 0.04; Kolmogorov–Smirnov criterion *p* = 0.01) compared to the group with low mitotic index Ki-67. This trend is observed both for adjacent noncancerous tissues and for the tissue of the peritumoral zone of the tumor; however, the differences with the low mitotic index Ki-67 group are not significant.

In the group with a high level of Ki-67 in nuclear protein, the content of glycogen synthase kinase 3β in the tissue of the peritumoral zone and glioma tumor was statistically significantly lower than in the group with a low mitotic index (Figure 6).

The content of glycogen synthase kinase 3β was significantly lower in the group with high mitotic index Ki-67 (U-criterion of Mann–Whitney *p* = 0.04; Kolmogorov–Smirnov criterion *p* = 0.02) compared to the group with low mitotic index Ki-67. This trend is observed both for adjacent noncancerous tissues and for the tissue of the peritumoral zone of the tumor; however, the differences with the low mitotic index Ki-67 group are not significant.

At a high level of Ki-67, the glucose level in tumor tissue was lower than in the group with a low level of this mitotic index (Figure 7).

Glucose concentration was significantly lower in the group with high mitotic index Ki-67 (U-criterion of Mann–Whitney *p* = 0.006; Kolmogorov–Smirnov criterion *p* = 0.02) compared to the group with low mitotic index Ki-67. This trend is observed both for adjacent noncancerous tissues and for the tissue of the peritumoral zone of the tumor; however, the differences with the low mitotic index Ki-67 group are not significant.

The differences in the activity of hexokinase in groups depending on the immunohistochemical profile of IDH1, p53, and Ki-67 were not detected. The activity of hexokinase was significantly different only depending on the methylation of gene promoter MGMT. In the group with a methylated promoter (the enzyme is not expressed), the activity of hexokinase in the peritumoral zone was significantly lower (Figure 8). Therefore, the greater the sensitivity to the action of chemotherapeutic preparation with an alkylating agent, the lower the activity of glycolysis in the peritumoral zone.

In the group with MGMT methylation, hexokinase activity in the peritumoral zone was significantly lower (U-criterion of Mann–Whitney *p* = 0.05; Kolmogorov–Smirnov criterion *p* = 0.05) than in the absence of methylation of the MGMT gene promoter.

Multidirectional changes in the activity of transketolase and hexokinase in the peritumoral zone of glial tumors (Figure 3 and Figure 8) were revealed in groups with a pronounced IDH1 mutation (significant increase) and with methylation of the MGMT gene promoter (significant decrease). In large samples, there is usually a correlation between IDH and MGMT, since the IDH mutation is often associated with the hypermethylation phenotype [23]. However, in this case, we are talking about the activity of enzymes in various metabolic processes: transketolase is an enzyme of the non-oxidative stage of the pentose phosphate pathway, and hexokinase is one of the regulatory enzymes of glycolysis. The severity of the activity of glycolysis and the pentose phosphate pathway in the same tissue can be multidirectional.

## 4. Discussion

The greatest differences in the metabolism of carbohydrates, compared to adjacent noncancerous tissues, were noted in the tumor tissue: the decrease in the concentration of lactate and the decrease in the level of glycogen synthase kinase-3β. For the peritumoral zone, no statistically significant differences with adjacent noncancerous tissues were found. Comparing the carbohydrate metabolism in the glioma tumor tissue with adjacent noncancerous tissues, it can be stated that aerobic glycolysis is activated in the tumor. There is a tendency to increase the rate of the oxidative stage of the pentose phosphate pathway and to decrease the intensity of its non-oxidative stage. The results obtained are consistent with the available data [24] on the multidirectional expression profile of glycolytic enzymes and enzymes of the pentose phosphate pathway in highly proliferative and highly migrating cells of gliomas.

IDH1 mutations are accompanied by higher activity of transketolase in the peritumoral zone which indicates the activation of the non-oxidative stage of the pentose phosphate pathway. IDH1 is located in the cytoplasm. Wild-type isoforms catalyze the oxidative decarboxylation of isocitrate to α-ketoglutarate with the formation of the reduced NADPH_2_. Mutant IDH1 and IDH2 catalyze the conversion of α-ketoglutarate into oncometabolite 2-hydroxyglutarate with the formation of oxidized NADP^+^ [25]. The direct relationship between the activity of transketolase in the peritumoral zone of the tumor and IDH1 was found with the local clustering coefficient between IDH1 and transketolase −0.637. A statistically significant increase in the activity of this enzyme in the peritumoral zone in the event of an IDH1 mutation is consistent with the available data on the normalization of glucose metabolism in gliomas with an IDH1 mutation which leads to slower progression of the tumor [26]. IDH mutations reduce glucose oxidation due to the inhibitory phosphorylation of pyruvate dehydrogenase [27]. In addition, glioma cells with IDH1 mutation demonstrate the activation of pyruvate carboxylase [28] which leads to the increase in the formation of oxaloacetate products. It is known [29] that glial tumor cells with an IDH1 mutation adjust the production of energy in the Krebs cycle due to the super-expression of LDH1.2 (with the formation of pyruvate). As a result, the pH of the tissue is normalized which explains the less aggressive biological behavior of gliomas with an IDH1 mutation. The decrease in the activity of lactate dehydrogenase 4.5 in the tissues of gliomas with an IDH1 mutation leads to the decrease in anaerobic glycolysis indirectly activating the pentose phosphate pathway. On the other hand, the possible coupling of the pentose phosphate pathway with the mutant form IDH1 is manifested at the NADP^+^ level which is formed in the reaction catalyzed by this enzyme (Figure 9).

Thus, an IDH1 mutation leads not only to the accumulation of 2-hydroxy glutarate, which affects the level of the subunits of HIF-1α factor induced by hypoxia, but also to significant changes in carbohydrate metabolism.

The decrease found in the activity of glucose-6-phosphate dehydrogenase in tumors tissue with expression p53, high local clustering coefficient between these proteins (0.766), and a significant inverse correlation relationship between them is explained by the fact that p53 protein is associated with glucose-6-phosphate dehydrogenase and prevents the formation of active dimer [30].

With a high mitotic index of Ki-67 in the tumor tissue, the level of glucose, transketolase, and glycogen synthase kinase 3β is reduced. Since the content of this tumor marker demonstrates the severity of the proliferative activity of the tissue, it can be stated that the mitotic activity of the tumor is associated with the metabolism of carbohydrates.

It is known that three main pathways of cell signaling exist being involved in the pathogenesis of gliomas [31], and glycogen synthase kinase 3 β is related to each of them (Figure 9). Thus, serine–threonine kinase Akt and mTOR (participants in the phosphoinositide-3-kinase/AKT/mTOR signal pathway) inhibit GSK3β [32]. On the other hand, GSK3β inhibits the insulin receptor substrate IRS which indirectly leads to inhibiting the PI3K pathway [33]. Another pathway, frequently modified by glial tumors, is the Ras pathway associated with the retinoblastoma protein RB [34]. Glycogen synthase kinase-3β reduces the amount of Cyclin D/CDK4 due to destabilization, adjusting the proliferation of the RAS signal pathway [35,36].

Apoptosis in gliomas is regulated through the ARF-MDM2-p53 pathway. In human gliomas, TP53 mutations or amplification of MDM2 [37] are possible. Glycogen synthase kinase 3β activates p53 both directly and via MDM2 [38].

A significant reduction in the level of glycogens synthase kinase-3β in the tumor tissue and the presence of significant inverse correlation bonds of this enzyme in the peritumoral zone and tissue of glial tumors with the mitotic index of Ki-67 demonstrates the importance of this enzyme for the invasive tumor activity. In addition to the relationship with the signaling paths being responsible for proliferation and apoptosis, GSK-3β adjusts the carbohydrate metabolism. It is known that GSK-3β phosphorylates and inhibits glycogen synthase repressing the synthesis of glycogen [39].

During methylation of the MGMT promoter, the hexokinase activity in the peritumoral zone decreased. The absence of MGMT activity leads to a decrease in the ability of tumor cells to restore the damaged DNA sections after the action of chemotherapeutic medication with alkylating agent [40,41,42]. That is, with greater sensitivity to the action of chemotherapy with an alkylating agent, the activity of glycolysis in the peritumoral zone is lower than in adjacent noncancerous tissues.

## 5. Conclusions

Thus, it was observed that the enzymes of glycolysis, of the pentose phosphate pathway, and the regulatory enzyme of carbohydrate metabolism of glycogen synthase kinase 3 β change their activity depending on the immunohistochemical profile of gliomas. Moreover, the level of nuclear protein Ki-67 has the greatest impact on carbohydrate metabolism. The identified interconnection of the processes of glycolysis with methylation of MGMT promoter makes it possible to use special features of glioma carbohydrate metabolism to improve the efficiency of chemotherapy.

## Figures and Tables

**Figure 1 biomedicines-10-01007-f001:**
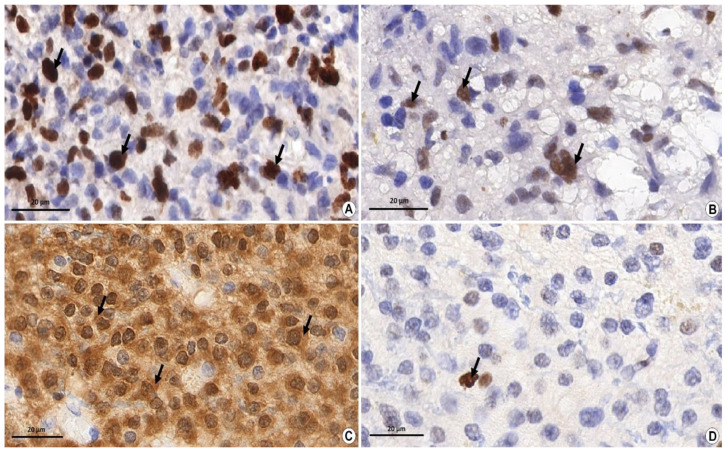
Immunohistochemical study of glial tumor markers. Magnification ×1000, Bar: 20 µm. (**A**) The level of nuclear non-histone protein—the marker of cell proliferation Ki-67 in the material of glioblastoma (grade IV). The diffusive pronounced nuclear brown staining (arrows) shows the binding reaction of antigen–antibody (up to 50% of cells in hot spots) which indicates a high proliferation index of Ki-67. (**B**) The presence of a p53 marker in the fragment of the anaplastic astrocytoma (grade III). Diffusive, moderately pronounced nuclear staining (arrows) shows the antigen–antibody binding reaction which indicates the presence of mutant protein p53. (**C**) The presence of an IDH1 mutation in the fragment of the diffusive astrocytoma (grade II). Diffusive, moderately pronounced brown staining in the cytoplasm shows the antigen–antibody binding reaction (arrows) which indicates the presence of an IDH1 mutation. (**D**) The presence of MGMT methylation in the material of the diffusive astrocytoma (grade II). Pinpoint (less than 10% of cells) nuclear brown staining (arrow) shows the antigen–antibody binding reaction which indicates the presence of methylation of the MGMT promoter.

**Figure 2 biomedicines-10-01007-f002:**
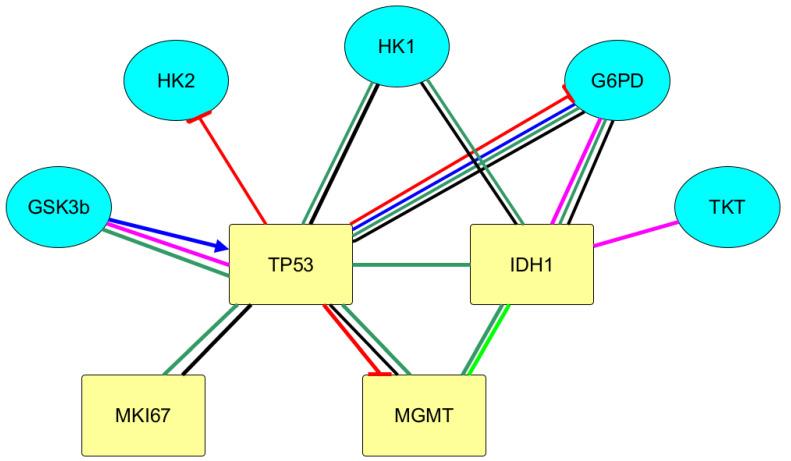
Integral scheme of protein–protein interactions of immunohistochemical tumor markers of gliomas with carbohydrates metabolism according to the databases of BioGrid, String, and Signor using the yEd Graph Editor program. Legend: GSK3B—glycogen synthase kinase 3 beta; TKT—transketolase; HK1—hexokinase 1; HK2—hexokinase 2; G6PD—glucose-6-phosphate dehydrogenase; IDH1—isocitrate dehydrogenase 1; MGMT—O-6-methylguanine-DNA methyltransferase; MKI67—marker of proliferation Ki-67; TP53—tumor protein p53.

**Figure 3 biomedicines-10-01007-f003:**
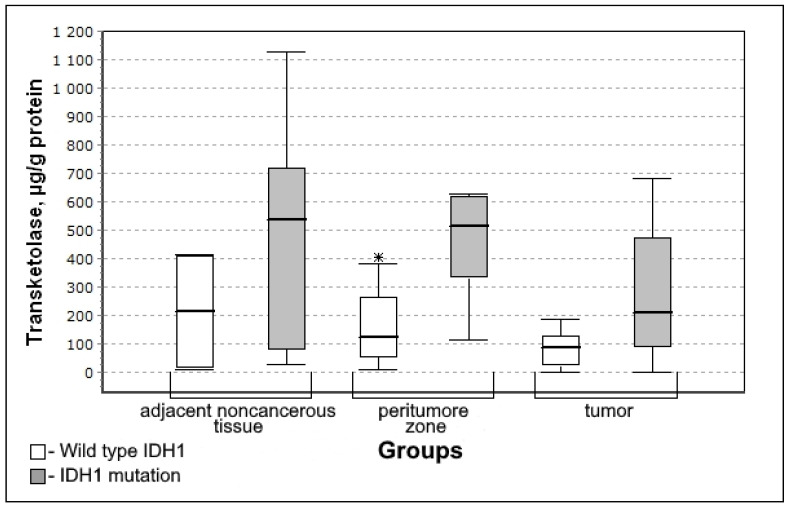
Medians and interquartile ranges (IQR) of the content of transketolase in the glioma tissue depending on the mutation of genes in iso-enzyme of the isocitrate dehydrogenase IDH1 (IDH1 mutation n = 12; wild type IDH1 n = 8). Statistics are reported * *p* < 0.05.

**Figure 4 biomedicines-10-01007-f004:**
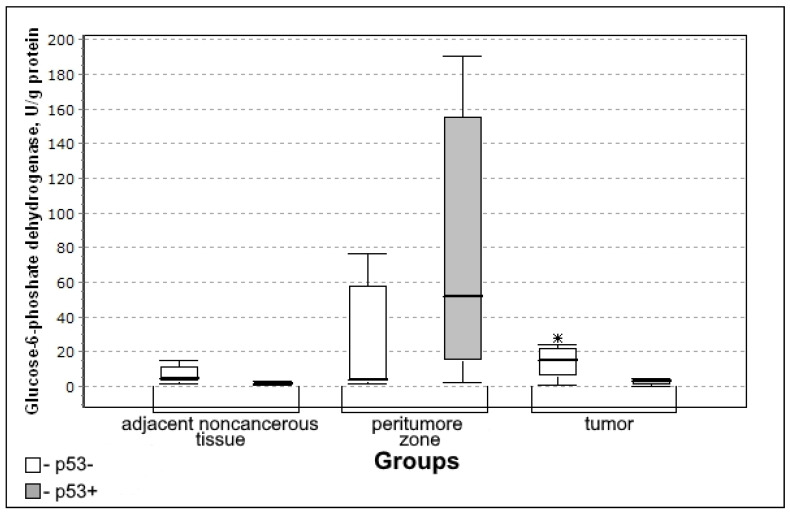
Medians and interquartile ranges (IQR) of the activity of the glucose-6-phosphate dehydrogenase in the tissue of gliomas depending on the expression of the tumor suppressor of p53 protein (p53 + n = 12; p53 − n = 8). Statistics are reported * *p* < 0.05.

**Figure 5 biomedicines-10-01007-f005:**
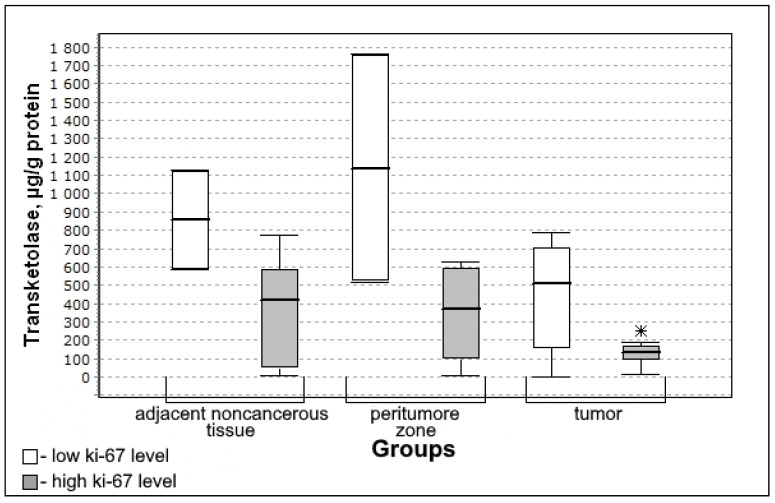
Medians and interquartile ranges (IQR) of the content of transketolase in the tissue of gliomas depending on the magnitude of the mitotic index Ki-67 (low Ki-67 n = 7; high Ki-67 n = 13). Statistics are reported * *p* < 0.05.

**Figure 6 biomedicines-10-01007-f006:**
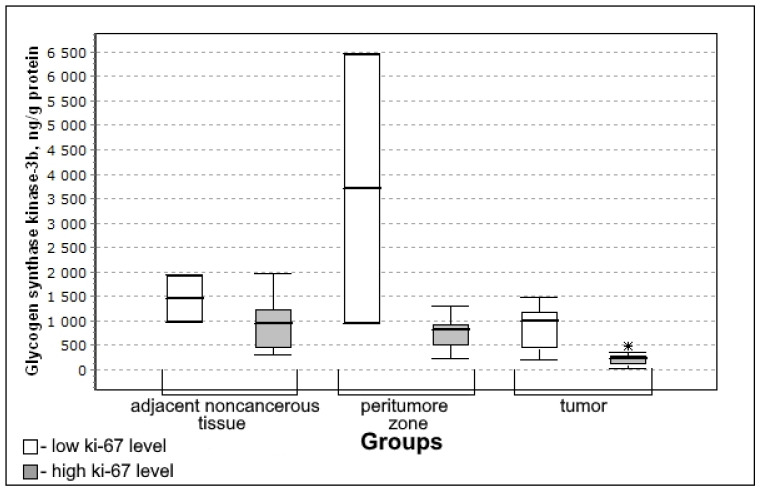
Medians and interquartile ranges (IQR) of the content of glycogen synthase kinase 3β in glioma tissue depending on the magnitude of the mitotic index Ki-67 (low Ki-67 n = 7; high Ki-67 n = 13). Statistics are reported * *p* < 0.05.

**Figure 7 biomedicines-10-01007-f007:**
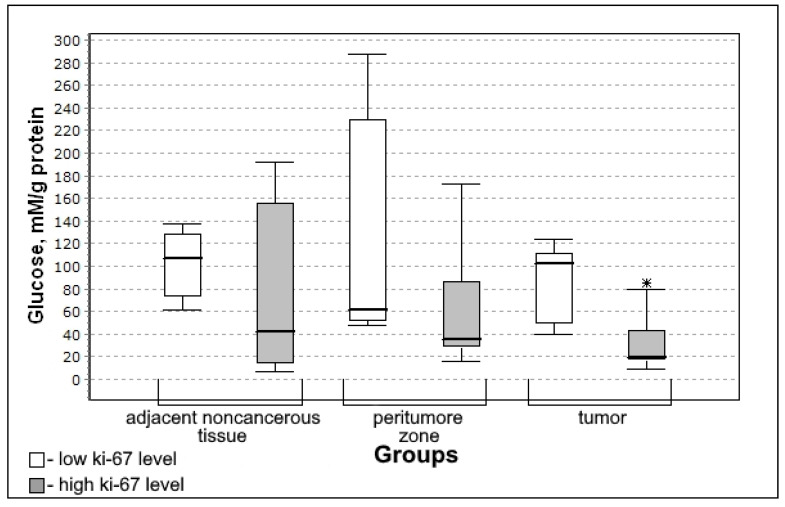
Medians and interquartile ranges (IQR) of concentrations of glucose in glioma tissue depending on the value of the mitotic index Ki-67 (low Ki-67 n = 7; high Ki-67 n = 13). Statistics are reported * *p* < 0.05.

**Figure 8 biomedicines-10-01007-f008:**
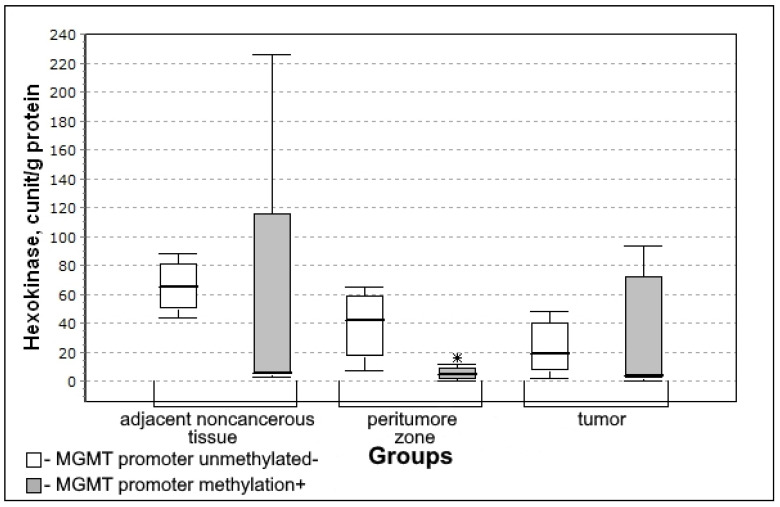
Medians and interquartile ranges (IQR) in the activity of hexokinase in glioma tissue depend on the promoter of the MGMT gene which carries out the DNA repair (MGMT promoter methylation n = 12; MGMT promoter unmethylated n = 8). Statistics are reported * *p* < 0.05.

**Figure 9 biomedicines-10-01007-f009:**
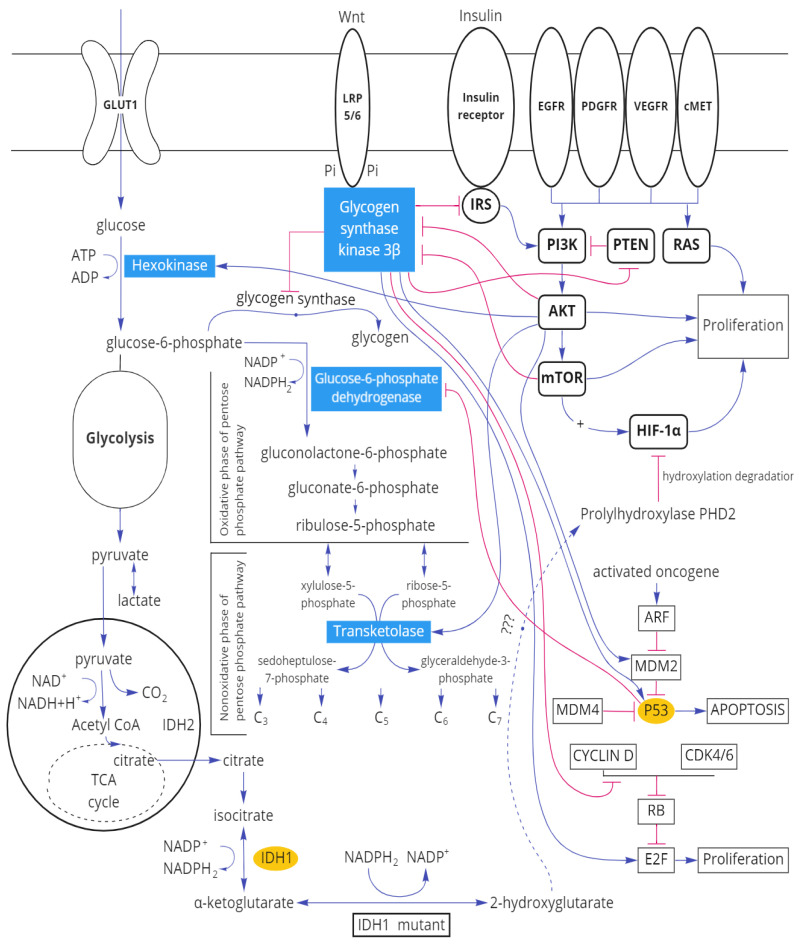
Scheme of the relationship of molecular tumor markers of gliomas with the main enzymes of carbohydrate metabolism.

**Table 1 biomedicines-10-01007-t001:** Morphometric estimation of tumor markers in glioma tissue with varying degrees of anaplasia in 20 individuals.

Grade	I	II	III	IV
Number of patients in group	1	6	3	10
IDH1—% of mutation presence	0%	100%	100%	30%
MGMT—% of methylation found	100%	83%	100%	30%
p53—% of presence	0%	66%	0%	80%
Ki-67 high (more than 10%)—% of presence	0%	0%	100%	100%
Ki-67 low (less than 10%)—% of presence	100%	100%	0%	0%
Median Ki-67	-	2%	12%	37%
Interquartile range of Ki-67 (25–75)	-	1.5–3%	-	25–40%

**Table 2 biomedicines-10-01007-t002:** Indicators of carbohydrate metabolism in gliomas.

	The Adjacent Noncancerous Tissues (Median; Quartiles)	Peritumoral Zone		Tumor	
		Median; Quartiles	U-Criterion of Mann–Whitney; Kolmogorov–Smirnov Criterion	Median; Quartiles	U-Criterion of Mann–Whitney; Kolmogorov–Smirnov Criterion
Glucose, mmol/g of protein	94.08 (44.36–164.84)	56.01 (38.53–163.89)	0.9181; 0.9738	47.81 (24.10–101.46)	0.1195; 0.1359
Lactate, mmol/g of protein	2.07 (1.02–4.43)	1.56 (0.36–4.67)	0.7576; 0.7502	0.49 * (0.15–1.53)	0.0942; 0.0299
Hexokinase, milliunit/mL/g of protein	6.21 (5.84–54.90)	8.93 (3.24–38.10)	0.8370; 0.5888	12.56 (1.63–37.35)	0.5463; 0.4360
Glucose-6-phosphate dehydrogenase, U/L/g of protein	4.59 (1.67–16.44)	11.76 (2.87–29.19)	0.3036; 0.7503	10.78 (2.88–28.44)	0.6128; 0.9737
Transketolase, µg/g of protein	266.44 (45.92–549.63)	237.08 (87.74–480.29)	0.9590; 0.7503	145.37 (57.83–366.79)	0.3305; 0.3684
Glycogen synthase kinase-3β, pg/mL/g of protein	998.17 (513.51–1944.6)	918.91 (725.92–1218.97)	0.7576; 0.8011	347.57 * (211.10–745.30)	0.0323; 0.0091

Legend: *—statistically significant differences in comparison with the adjacent noncancerous tissues of the brain.

**Table 3 biomedicines-10-01007-t003:** Correlations (the Spearman’s coefficient) between immunohistochemical markers of gliomas and parameters of carbohydrate metabolism of brain tumors in 20 individuals.

Parameter of Carbohydrate Metabolism		Markers of Gliomas			
		IDH	Ki-67	MGMT	p53
Glucose Rho (*p*)	Adjacent noncancerous tissue	−0.194 (0.568)	0.234 (0.544)	−0.504 (0.203)	0.109 (0.797)
	Peritumoral zone	−0.211 (0.488)	0.012 (0.973)	−0.142 (0.695)	0.0001(0.993)
	Tumor tissue	0.073 (0.780)	−0.558 * (0.031)	0.163 (0.595)	0.116 (0.735)
Lactate Rho (*p*)	Adjacent noncancerous tissue	0.065 (0.851)	−0.042 (0.915)	−0.630 * (0.044)	0.109 (0.797)
	Peritumoral zone	0.127 (0.680)	−0.055 (0.881)	0.0001 (1)	0.109 (0.797)
	Tumor tissue	0.122 (0.641)	−0.189 (0.501)	−0.096 (0.756)	0.346 (0.297)
Glycogen synthase kinase-3β Rho (*p*)	Adjacent noncancerous tissue	0.194 (0.568)	−0.201 (0.604)	−0.630 * (0.044)	0.218 (0.604)
	Peritumoral zone	0.296 (0.327)	−0.407 (0.243)	0.071 (0.845)	0.218 (0.604)
	Tumor tissue	0.024 (0.926)	−0.417 (0.123)	0.067 (0.828)	−0.116 (0.735)
Glucose-6-phosphate dehydrogenase Rho (*p*)	Adjacent noncancerous tissue	−0.194 (0.568)	−0.167 (0.667)	0.0001 (0.974)	−0.546 (0.162)
	Peritumoral zone	−0.127 (0.680)	0.480 (0.160)	−0.497 (0.144)	0.109 (0.797)
	Tumor tissue	−0.366 (0.149)	0.294 (0.287)	−0.278 (0.359)	−0.635 (0.036) *
Hexokinase Rho (*p*)	Adjacent noncancerous tissue	−0.323 (0.333)	0.209 (0.589)	−0.474 (0.282)	0.0001(0.968)
	Peritumoral zone	−0.085 (0.784)	0.219 (0.544)	−0.693 (0.039) *	−0.109 (0.797)
	Tumor tissue	−0.293 (0.254)	0.058 (0.866)	0.056 (0.863)	0.058 (0.866)
Transketolase Rho (*p*)	Adjacent noncancerous tissue	0.281 (0.443)	−0.2845 (0.458)	0.0001 (0.946)	0.109 (0.797)
	Peritumoral zone	0.507 (0.077)	−0.164 (0.651)	−0.142 (0.695)	0.218 (0.604)
	Tumor tissue	0.268 (0.298)	−0.368 (0.177)	−0.057 (0.852)	−0.404 (0.218)

Legend: Statistics are reported * *p* < 0.05.

## Data Availability

The datasets generated during and/or analyzed during the current study are available from the corresponding author on reasonable request.

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
