# Peer review of "Carbohydrate Metabolism Parameters of Adult Glial Neoplasms According to Immunohistochemical Profile"

_biomedicines, 2022, doi:10.3390/biomedicines10051007_

Round 1

Reviewer 1 Report

The authors investigated the relationship between carbohydrate metabolism
parameters and immunohistochemical features of gliomas, by using tissue from tumor, peritumoral area and adjacent unaffected brain (20 patients in total). Decreased levels of lactate and glycogen synthase kinase-3β were found in the tumor tissue with respect to unaffected adjacent tissue. The activity of the carbohydrate metabolism enzymes was closely related to the immunohistochemical glioma profile, especially Ki-67 level. The authors also emphasized that this relationship between glycolysis with methylation of the promoter of O-6-methylguanine-DNA-methyltransferase (MGMT) of gliomas may be used to increase chemotherapy efficiency. 

The paper is very interesting and well written. The experiments well performed and the results also well explained.

I have some suggestions to improve the manuscript:

  1. The comparison between tumor tissue and peritumoral areas is one of the most debated topic in neuro-oncology and neuropathology. The authors must improve the description of the "status of the art" about this interesting topic. The following doi should be used: 10.3390/brainsci11020200
  2. Did the authors perform additional glioma prognostic factors such as SRSF1 for example? 

Author Response

Dear reviewer!

We would like to thank you for the positive assessment of our work and meaningful suggestions for its improvement!

  • We have reviewed the style and spelling of the work once again and made corrections accordingly
  • We studied in-depth the data of the offered article doi: 10.3390/brainsci11020200 and used it in the Introduction. Thank you!
  • The definition of SRSF1 was not planned within the context of this research. Regarding a number of neoplasms, we conducted an additional study in order to determine the status of VEGF, however due to a small number of cases with the definite status of vascular factor, representativeness of data in this line of research was considered to be too low.
  • The list of references was increased by 2 articles, so the numbering of sources was changed in the text

With best regards, authors of the article

Reviewer 2 Report

The authors found that the enzymes of glycolysis, of pentose phosphate pathway and the regulatory enzyme of carbohydrate metabolism of glycogen synthase kinase 3β change their activity depending on the immunohistochemical profile of gliomas. Moreover, the level of nuclear protein Ki-67 has the greatest impact on carbohydrate metabolism. The identified interconnection of the processes of glycolysis with methylation of MGMT promoter makes it possible to use special features of glioma carbohydrate metabolism to improve the efficiency of chemotherapy.

This is a very interesting study and manuscript was well written. The conclusion was supported by the experimental data. I have some minor concerns.

  1. Regarding Figures 3 and 8, there are big variation in some groups in these two figures. The author should explain it in the discussion and also what you should do in the future for this big variation.
  2. For figures 3 to 8, the group number should be listed in the figure legend.
  3. IDH first appears should be listed as full name.
  4. Some extra spaces in some sentences were found in the manuscript, please remove it and pay attention to it during your resubmission.

Author Response

Dear reviewer!

We would like to thank you for the positive assessment of our work and meaningful suggestions for its improvement!

  • We explained differences in Fig. 3 and 8 at the end of the demonstration of results.
  • In accordance with your comment (Thank you!) we made corrections in the caption to Fig. 3-8.
  • In accordance with your comment the first reference of IDH was made with the full name.
  • All extra spaces were deleted, these locations were highlighted in yellow.
  • The list of references was increased by 2 articles, so the numbering of sources was changed in the text

With best regards, authors of the article

Reviewer 3 Report

This manuscript studied the changes of carbohydrate metabolism parameters between glioma tissue and adjacent nontumoral tissue as well as peritumor zone. Lactate and glycogen synthase kinase-3ß were found significant reduction in the glioma tissue. Different carbohydrate metabolism enzyme activity correlated to several glioma pathology markers: IDH1, p53, ki-67 and MGMT. The data may provide insights for mechanistic dissection of differential response to drug therapy targeting on carbohydrate metabolism and MGMT. 

1. Table 3 can be better presented to see each parameter difference side by side of noncancerous tissue, peritumoral zone and tumor tissue. 

2. Experimental evidence needs to provide for validation of protein-protein interaction discovered with bioinformatic analysis in Figure 2.    

Author Response

Dear reviewer!

We would like to thank you for the positive assessment of our work and meaningful suggestions for its improvement!

  • We have reviewed the style and spelling of the work once again and made corrections accordingly
  • In accordance with your comment (Thank you!) we made corrections in the Table 3.
  • Our experimental evidence partially to provide for validation of protein-protein interaction discovered with bioinformatic analysis in Figure 2. We attribute this to the small size of data sampling.
  • The list of references was increased by 2 articles, so the numbering of sources was changed in the text

With best regards, authors of the article